# Patterns and Driving Factors of Diversity in the Shrub Community in Central and Southern China

**Nan Deng** [1,2] , **Qingan Song** [1,2], **Fengfeng Ma** [1,2] and **Yuxin Tian** [1,2,*]

1   Hunan Academy of Forestry, No. 658 Shaoshan Road, Changsha 410004, China; dengnan@hnlky.cn (N.D.); qingansong@163.com (Q.S.); mafengfeng0403@126.com (F.M.)
2   Hunan Cili Forest Ecosystem State Research Station, Changsha 410004, China
*   Correspondence: tianyuxineco@163.com

**Abstract:** Climate, topography, and human activities are known to influence plant diversity. In the present study, species-abundance distribution (SAD) patterns of the shrub community were fitted, and the mechanism of contribution of 22 driving factors was assessed. The results showed that the $\alpha$-diversity index exhibited no significant differences between artificial disturbance and the natural community. The Zipf and Zipf–Mandelbrot models were found to exhibit a good SAD fitting of the communities, thereby exhibiting a different diversity structure. It was observed that the SAD followed more than one rule, and the Zipf–Mandelbrot model was better than other models. The gradient boosting model indicated that precipitation in the wettest month, annual precipitation, and slope direction showed the strongest impact on plant richness. The indicator species of the artificial disturbance and natural community were identified from a multiple regression tree. Furthermore, an increase in species diversity was observed with a rise in latitude, exhibiting a single-peaked curve with increased altitude. $\beta$-diversity analysis indicated that both habitat filtering and the neutral effect influenced the establishment of the natural community, while the establishment of the artificial disturbance community was only affected by habitat filtering. Our study provides a better understanding of the ecological process of the maintenance of shrub-community diversity.

**Keywords:** understory vegetation diversity; species-abundance distribution; indicator species; species pool; afforestation; multiple regression tree

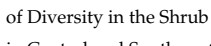



## 1. Introduction

Determination of species diversity is one of the most fundamental topics in biology and ecology [1,2] and plays a crucial role in biodiversity conservation, planning of nature reserves, and forest management [3]. The basic premise of modern ecology is to measure species abundance, which reflects the universality and rarity of the species and is used as the main judgment basis to ascertain the dominance and evenness of different species [4]. Species-abundance distribution (SAD) characterizes the structure of species abundance and is the core focus of macroecology [4,5]. A considerable effort has been made to characterize empirical SADs in a statistically tractable framework. Different SAD models, such as log-series- and log-normal-like shaped and mixed gamma binomial distribution, have been proposed to verify species assembly rules [6]. Comparison of the SAD models is often used to detect disturbance and damage to the ecosystem and explain the resource allocation and interspecies associations [7–9]. The diversity structure assists in describing and explaining current biodiversity patterns, and has been a focus of ecological studies over the last few decades. However, the anthropogenic and environmental factors exhibiting the strongest impact on forest diversity patterns remain unclear [10,11]. Previous studies have shown that the environmental effects of the other driving variables (such as climate, topographic heterogeneity, and soil factors) and human activities exhibit a substantial impact on the spatial distribution of species [12–14]. Climatic gradient is considered

the main abiotic factor controlling large-scale species diversity [15]. Some studies have shown that spatial patterns of tree species richness are mainly influenced by water and energy [16,17]. Topographic and soil factors exert significant effects on the species richness pattern [18,19]; however the influence of the topographic factors on plant richness has not been extensively studied. Further, the effect of the hump-type diversity of forests on a regional scale remains unclear. Meanwhile, there is a growing controversy focusing on the richness, composition, and survival of the biodiversity, given persistent anthropogenic disturbances [20]. Studies have shown that anthropogenic disturbance can reduce tropical forest biodiversity [21], while others have found that tropical forests were associated with poorer species diversity without disturbance [22,23].

Most studies on biodiversity patterns focus on species richness, but ignore the relative abundance of the species and its effect on interspecies interactions [24]. β-diversity refers to the variability among the communities at spatial and temporal scales in terms of species composition, evolutionary relationships, and functional attributes [25]. β-diversity may reflect the dynamic nature of the biodiversity patterns better than the simple measures of α-diversity alone [26]. Two major theories that explain the diversity gradients are the niche and neutral theories. The niche theory emphasizes the importance of the contemporary environment (habitat filter), while the neutral theory believes that community dynamics is a random process and that dispersal limitations play a role in the community structure [27]. Thus, analysis of patterns of different aspects of diversity on different scales is necessary to identify the ecological processes.

Shrubs have diverse ecological functions, and contribute to the forest carbon stock. The species assemblages may have different patterns of diversity than the tree species [28]. The area selected in the present study (Hunan province) is located in central–south China. It is located between 24°38′–30°08′ N and 108°47′–114°15′ E, and the total area is 211,800 km$^2$. The terrain is surrounded by mountains on three sides and is open to the north. It is composed of plains, basins, hills, mountains, rivers, and lakes. It crosses the Yangtze River and the Pearl River, and has a subtropical monsoon climate. The uneven distribution of the hydrothermal conditions form a highly heterogeneous habitat unit and vegetation distribution pattern in space. As an important component of the plant diversity in this area, the shrub community is often distributed in ecologically vulnerable areas. Part of this has been seriously disturbed by human activities; however, there is a lack of analysis of the diversity pattern exhibited by the shrub community. The current diversity pattern is considered to be a result of the concurrent effect of natural factors and anthropogenic disturbance; however, the effects of natural factors and anthropogenic disturbance on the pattern remain controversial. Additionally, the maintenance of shrub-community diversity has practical significance in the restoration of the damaged forest or subject to long-term human intervention. Therefore, the objective of this study was to explore the diversity pattern of the shrub community and compare the differences between the natural and artificial disturbance communities. The domain factors that affect the diversity pattern were studied. Furthermore, the relative importance of the habitat filters and dispersal limitation on β-diversity was revealed.

## 2. Materials and Methods

### 2.1. Data Sources

The location under study was an ecological forest in Hunan. An ecological forest has the function of maintaining ecological balance and protecting biodiversity. The ecological forest is the main forest resource and covers 36.65% of Hunan, and deforestation is not allowed. Forest survey data were obtained from the forest fixed sample plot investigation database of Hunan ecological forests (updated in 2019). A total of 683 fixed sample plots in this database were set at equal spaces in the area of ecological forest, and the space was calculated according to the total forest area using geographic information system (GIS) software. The size of each plot was 25 m (vertical to the counter line) × 40 m (parallel to the contour line). Investigation of all the plots was conducted in 2019, and the origin, forest

type, status, and degree of interference of human activities were recorded. Stand dynamics factors—altitude, soil type, and slope direction—of each plot were also studied. The sites of the shrub plots were selected from the database, and the forest with shrub coverage greater than or equal to 40% was identified as the shrub community. In this study, a total of 39 plots were selected (Figure 1). The formation of the shrub communities is known to be caused by natural regeneration and human activities. In this study, the plots were divided into two categories: natural (24) and artificial disturbance (15) communities. The natural community was formed due to natural regeneration (including regeneration from deforestation, natural disasters and so on) and was free from human interference, while the artificial disturbance community was formed due to the plantation of economic shrubs (tea-oil tree and citrus). The shrub species, their abundance, and stand dynamics factors (altitude, location information, soil type, and slope direction) were extracted. Additionally, 19 Bioclim variables of each site were retrieved from the world climate database (WorldClim: http://www.worldclim.org/, accessed on 1 December 2021, Table 1).

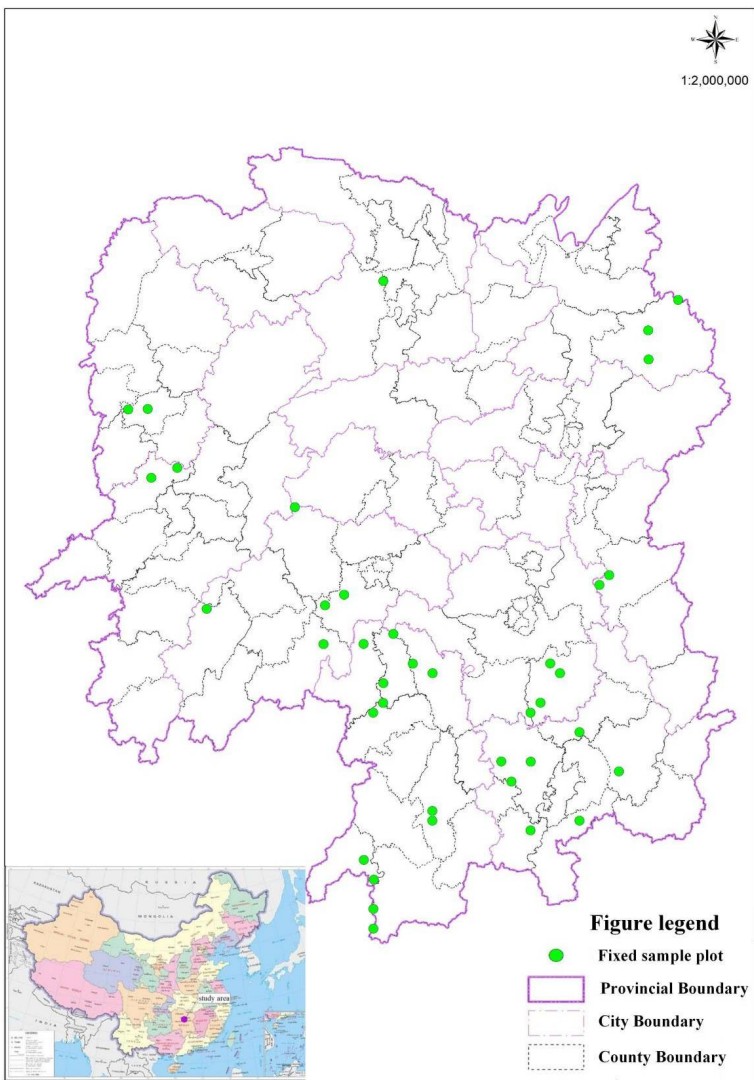

**Figure 1.** Location of the study area. Green circles represent the plots.

**Table 1.** Bioclim variables.

| Code | Description |
|---|---|
| Bio1 | Annual Mean Temperature |
| Bio2 | Mean Diurnal Range (Mean of monthly (max temp.–min temp.)) |
| Bio3 | Isothermality (BIO2/BIO7) (×100) |
| Bio4 | Temperature Seasonality (standard deviation ×100) |
| Bio5 | Max Temperature of Warmest Month |
| Bio6 | Min Temperature of Coldest Month |
| Bio7 | Temperature Annual Range (BIO5-BIO6) |
| Bio8 | Mean Temperature of Wettest Quarter |
| Bio9 | Mean Temperature of Driest Quarter |
| Bio10 | Mean Temperature of Warmest Quarter |
| Bio11 | Mean Temperature of Coldest Quarter |
| Bio12 | Annual Precipitation |
| Bio13 | Precipitation of Wettest Month |
| Bio14 | Precipitation of Driest Month |
| Bio15 | Precipitation Seasonality (Coefficient of Variation) |
| Bio16 | Precipitation of Wettest Quarter |
| Bio17 | Precipitation of Driest Quarter |
| Bio18 | Precipitation of Warmest Quarter |
| Bio19 | Precipitation of Coldest Quarter |

*2.2. Species Diversity and Fitting of SAD*

Species diversity ($\alpha$-diversity) was assessed by using the most commonly used diversity indices—richness, Shannon, Shannon entropy, Shannon's evenness (Hill ratio), Simpson, Simpson's evenness (Hill ratio), Pielou's evenness—and pairwise comparison was performed using the Wilcoxon test. Rank abundance dominance (RAD) plots were constructed to display the logarithmic species abundance against the species rank order, in order to analyze the types of abundance distributions. In this study, 5 models were used: broken stick, niche preemption, log normal model, Zipf, and Zipf–Mandelbrot [29,30]. The Akaike information criterion (AIC) and Bayesian information criterion (BIC) were used to compare the models, and smaller AIC and BIC values indicated a better fitting effect. The K-S test was used to test the models, and the D statistic of the two empirical distribution functions were calculated to identify significant differences.

The shape parameter of the gambin model [31] is an alternative approach that focuses on a single value to characterize the shape of the SAD [5]. Gambin is a stochastic model that combines the $\gamma$-distribution with a binomial sampling method, and the single free parameter ($\alpha$) characterizes the distribution shape. Low values indicate log-series-shaped curves and a higher proportion of rare species, whereas higher values indicate more log-normal-shaped curves [5]. We fitted the unimodal, bimodal, and trimodal versions of the gambin model to the data and then compared the three models using the BIC.

*2.3. Prediction of Species Pool and Co-Occurrence Network*

Species accumulation models indicate that not all species can be seen in any given site, and these unseen species also belong to the species pool. Three models were used in the prediction of the total number of species in the study area, i.e., Chao model [8,32], Jackknife model, and Bootstrap model [33]. The species number of each site was also estimated using the Chao and ACE models [8,34], and the probability of occurrence in each site was calculated using the Beal smoothing model [35]. In order to explore the relationship between the composition of the species and the gradient of environmental factors, a multiple regression tree (MRT) was used based on the Hellinger coefficient. Indicator species analysis (ISA) is an effective method to determine the response of a species to the environment, wherein the indicator value is calculated according to the species distribution among groups. The indicator-value indices of species were calculated in the range 0 to 1, where a higher value indicated the efficiency the indicator [36]. At the

same time, 999 Monte Carlo tests were performed. In each test, each site was randomly assigned to different groups, and the indicator value of each species in each group was recalculated [37]. Type I error (i.e., the probability that the highest indicator value of the same species in a random test equals or exceeds the highest indicator value of the actual species) was used to test the significance of the highest indicator value for each species [38]. A co-occurrence network was built to study the interactions between the species, and an undirected symbiotic network was constructed based on the Jaccard similarity matrix, and dense connected subsets (species sets with high symbiotic frequency, i.e., module) and their internal associations were detected.

### 2.4. Detection of Driving Factors in Community Construction

We quantified the relative contribution of 22 driving factors—19 Bioclim variables, soil type, altitude, and slope direction (mentioned above)—to study diversity with the gradient boosting model (GBM) [38] based on the Shannon index of each plot. This model continuously fits the nonlinear relationship between diversity and factors, and its flexibility, explanatory variable selection, and cross-validation approach offer an advantage in ecology studies [39,40]. Additionally, marginal plots were constructed that reflected the influence from one predictor variable when other predictor variables were fixed. Linear regression was carried out between the diversity index (richness, Shannon and Simpson diversity) and altitude/latitude to detect the horizontal distribution pattern of diversity. To detect the relationship between β-diversity (Bray–Curtis distance), environmental factors and geographic coordinates (latitude and longitude) were standardized and environmental and geographic Euclidean distance between the pairwise plots calculated. Mantel analysis was used to detect the relationship between the geographic distance matrix, environmental distance matrix and diversity matrix. Partial Mantel was used to study the explanatory quantity of environmental distance and geographic distance with the change in diversity.

### 3. Results

#### 3.1. Fitting of SAD

The seven α-diversity indices exhibited no significant differences between the natural and artificial disturbance communities, except for species richness ($p > 0.05$, Figure 2). The fitting results of the broken-stick, niche-preemption, log-normal, and Zipf–Mandelbrot models are shown in Figure 3 and Table 2. The K-S test result indicated that all the SAD models were acceptable ($p < 0.05$), thereby proving that the SAD of shrub communities followed both log-series- and log-normal-like shaped distributions. The overlap of the fitting curves of Zipf and Zipf–Mandelbrot with the shrub communities indicated a similar fitting effect. AIC and BIC indicated that the Zipf model was the best-fitting model for the shrub community, followed by the Zipf–Mandelbrot model. The Zipf–Mandelbrot model was selected for comparative analysis. Parameter 2 of the Zipf–Mandelbrot model yielded low values in highly organized systems with complex interactions among species. Parameter 3 represented the potential diversity of the environment or niche diversification, and had higher values when the environment provided room for more alternatives [41]. It was found that parameter 2 of the Zipf–Mandelbrot model was −0.71 (whole community), −0.63 (artificial disturbance community), and −0.64 (natural community), which indicated that the organizational structure of the natural community was more balanced. Parameter 3 was 0.81 (whole community), $3.37 \times 10^{-5}$ (artificial disturbance community), and 0.67 (natural community), indicating that the dominance of the dominant species of artificial disturbance community was very high compared to that of the natural community.

Additionally, the unimodal, bimodal, and trimodal gambin models were applied to fit SAD. The unimodal gambin model provided the best fit to the shrub community (BIC = 247.79, 80.83 and 173.21; Figure 4), followed by the trimodal and bimodal gambin models. Parameter α of the unimodal gambin model was 0.54 (whole community), 0.61, (artificial disturbance community) and 0.53 (natural community). A higher α-value indicated strong diffusion restrictions in a community and that there were many rare species

that could easily disappear or were on the verge of extinction in the community. The results suggested a weak diffusion limit in the artificial disturbance community, thereby suggesting that the individuals of the community were mostly immigrants.

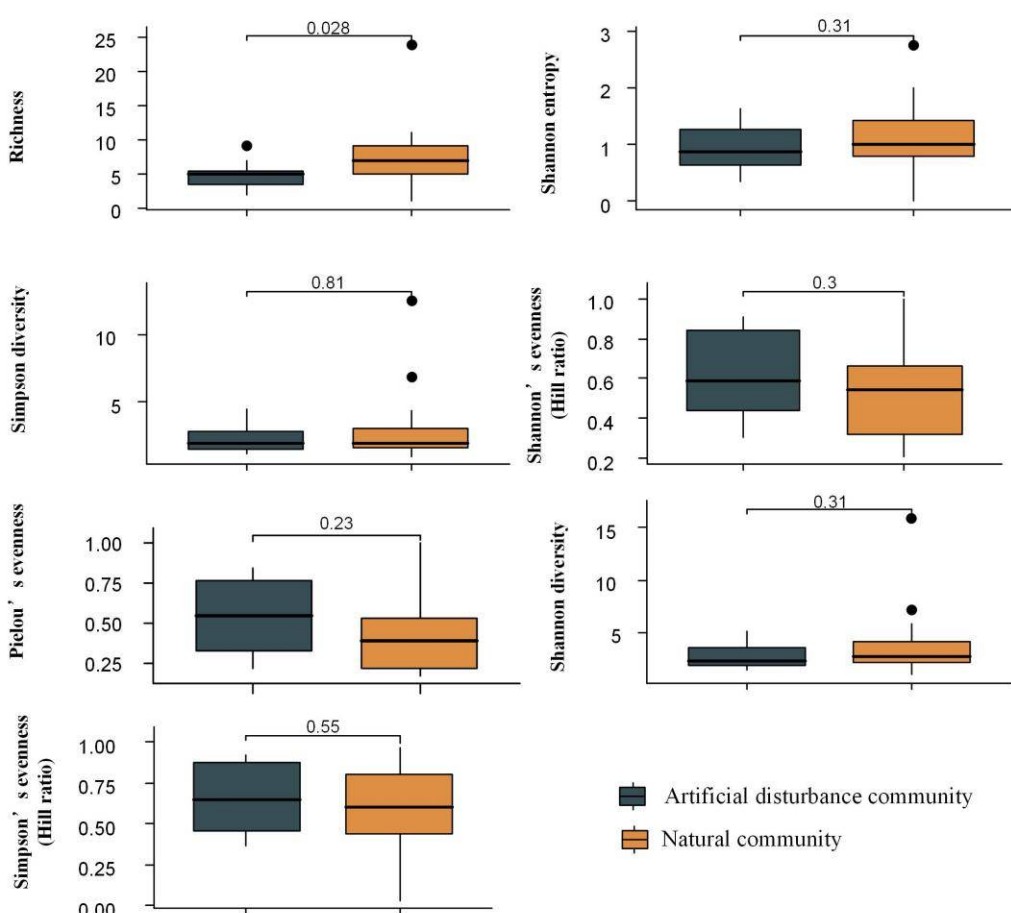

**Figure 2.** Comparison of diversity indices between the natural and artificial disturbance community (numbers above the bars indicate *p*-value).

**Table 2.** Fitting results of five SAD models.

| Type | All Plots | | | | |
|---|---|---|---|---|---|
| Model | M1 | M2 | M3 | M4 | M5 |
| Parameter 1 | / | 0.021363 | 0.38183 | 0.073054 | 0.096087 |
| Parameter 2 | / | / | 0.76961 | −0.63648 | −0.70504 |
| Parameter 3 | / | / | / | / | 0.80859 |
| Deviance | 69.0791 | 47.9878 | 24.3118 | 6.2648 | 5.5792 |
| AIC | 362.9997 | 343.9084 | 322.2324 | 304.1854 | 305.4998 |
| BIC | 362.9997 | 346.7287 | 327.873 | 309.8259 | 313.9607 |
| D statistic | 0.39516 | 0.37903 | 0.32258 | 0.3629 | 0.32258 |
| *p*-value of K-S test | $7.8 \times 10^{-9}$ | $3.67 \times 10^{-8}$ | $4.98 \times 10^{-6}$ | $1.62 \times 10^{-7}$ | $4.98 \times 10^{-6}$ |
| Type | Artificial disturbance | | | | |
| Model | M1 | M2 | M3 | M4 | M5 |
| Parameter 1 | / | 0.05423 | 0.33715 | 0.11671 | 0.11671 |
| Parameter 2 | / | / | 0.64207 | −0.63031 | −0.63032 |
| Parameter 3 | / | / | / | / | $3.37 \times 10^{-5}$ |
| Deviance | 21.8428 | 11.1016 | 6.5021 | 1.7376 | 1.7376 |
| AIC | 116.4851 | 107.7438 | 105.1444 | 100.3798 | 102.3798 |

**Table 2.** *Cont.*

| Type | Artificial disturbance | | | | |
|---|---|---|---|---|---|
| BIC | 116.4851 | 109.4574 | 108.5715 | 103.807 | 107.5206 |
| D statistic | 0.43902 | 0.41463 | 0.36585 | 0.34146 | 0.34146 |
| *p*-value of K-S test | 0.0007397 | 0.001737 | 0.008274 | 0.01678 | 0.01678 |
| Type | Natural | | | | |
| Model | M1 | M2 | M3 | M4 | M5 |
| Parameter 1 | / | 0.022811 | 0.32028 | 0.068435 | 0.084815 |
| Parameter 2 | / | / | 0.65743 | −0.58301 | −0.63843 |
| Parameter 3 | / | / | / | / | 0.66683 |
| Deviance | 60.4249 | 32.4401 | 18.8711 | 5.8502 | 5.5021 |
| AIC | 292.1572 | 266.1724 | 254.6034 | 241.5825 | 243.2344 |
| BIC | 292.1572 | 268.7876 | 259.8337 | 246.8127 | 251.0798 |
| D statistic | 0.44554 | 0.40594 | 0.36634 | 0.35644 | 0.35644 |
| *p*-value of K-S test | $3.92 \times 10^{-9}$ | $1.18 \times 10^{-7}$ | $2.60 \times 10^{-6}$ | $5.35 \times 10^{-6}$ | $5.35 \times 10^{-6}$ |

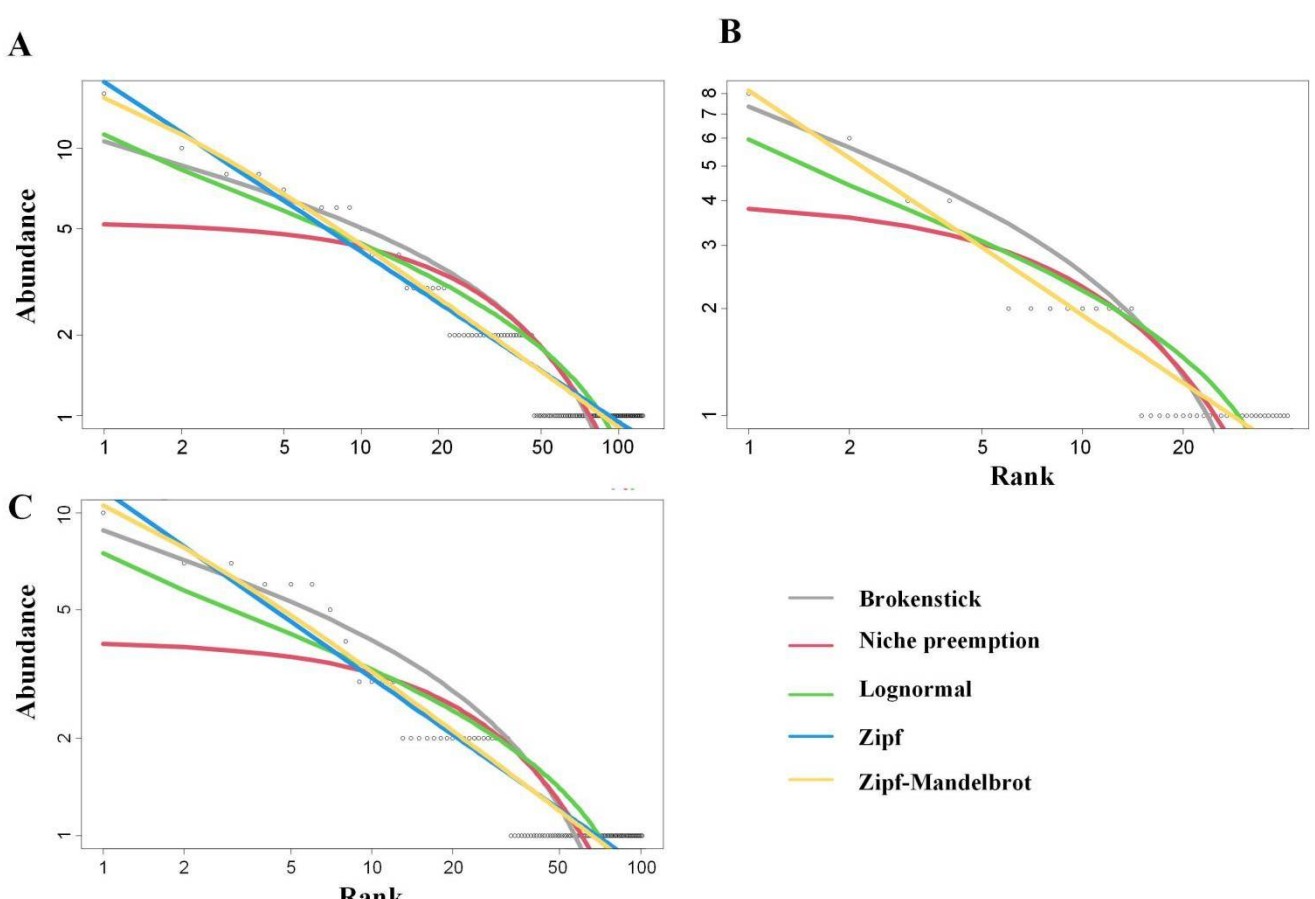

**Figure 3.** Fitting plots of different SAD models. (**A–C**) represents all, artificial disturbance, and natural plots (M1–M5 represent broken stick, niche preemption, log normal, Zipf, and Zipf–Mandelbrot).

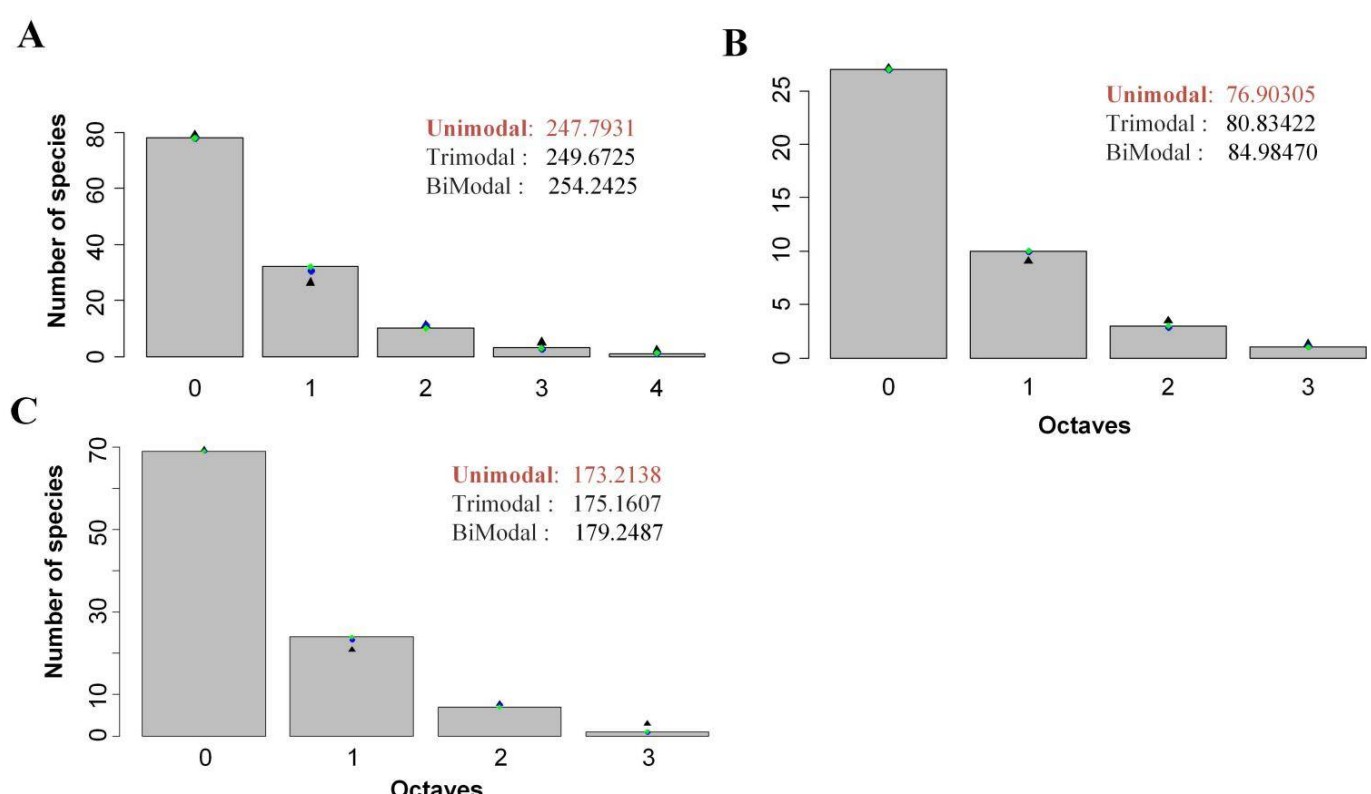

**Figure 4.** Fitting plots of the unimodal (black), bimodal (blue), and trimodal (green) gambin models. (**A–C**) represent the all, artificial disturbance and natural plots.

### 3.2. Species Pool and Co-Occurrence Network

Species accumulation curves showed that there were still some species that were not observed or underestimated in the distribution of the existing species (Supplementary Figure S1). Three models were used to estimate the species richness of the shrub community (Table 3). The results indicated that the richness predicted by the Chao model was the highest, followed by the jackknife and bootstrap models, respectively. The predicted richness was 243 (Chao), 200 (jackknife), and 155 (bootstrap). Additionally, the species richness in each plot was also calculated using the Chao and ACE models (Supplementary Table S1). The observed species richness in most sample plots (over 80%) accounted for more than 60% of the predicted species richness, suggesting that most species in the plot were sampled. The occurrence of the probability for each species in each plot was also predicted using the Beal smoothing model (Supplementary Figure S2). The co-occurrence network of the natural and artificial disturbance community are presented in Supplementary Figure S3. It was observed that six modules were detected in the artificial disturbance community, and the inter-module exhibited a close connection. Many modules were detected in the natural community, and there was little connection between the inter-modules.

**Table 3.** Predicted results of the three models.

| Model | Predicted Value | Variance |
| --- | --- | --- |
| Observed species richness | 124 | - |
| Chao model | 243 | 37.44 |
| Jackknife model | 200 | 20.75 |
| Bootstrap model | 155 | 10.32 |

### 3.3. Driving Factors of Diversity

The importance of six factors is presented in Figure 5A, including bio 13 (precipitation of wettest month, over 35%), altitude (over 20%), bio 12 (annual precipitation, over 15%), slope, and so on. The effect of the most important four factors on diversity showed that diversity sharply increased when bio 13 and altitude reached about 230 mm and 600 m, respectively, and remained unchanged. The diversity sharply decreased when bio 12 reached about 1380 mm and remained unchanged. The plots in the E-N, E-S, W, and W-N slopes exhibited more higher diversity (Figure 5B). Additionally, a two-way marginal analysis was conducted (Figure 5C), and the results showed that diversity reached a maximum value when bio 13 was below 225 mm and the altitude over 250 m. The three-way margin plot of bio 13, altitude, and slope showed the same trend as the two-way marginal plot in all slope directions (Figure 5D).

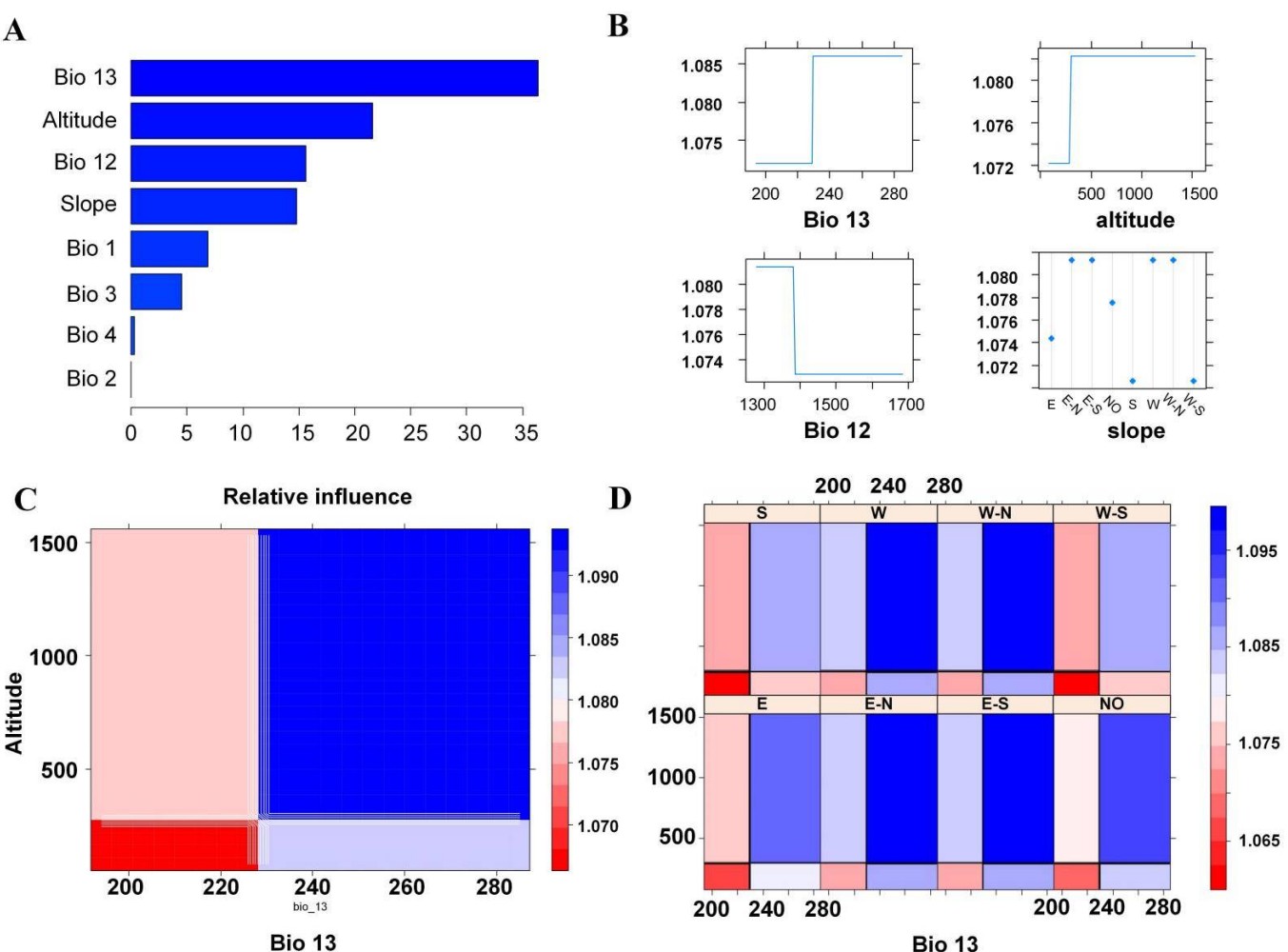

**Figure 5.** Results of GBM. (**A**) Importance of single factors, (**B**) effect of most important four factors on the diversity, (**C**) marginal effect of bio 13 and altitude on diversity, (**D**) three-way (bio 13, altitude, and slope) marginal effect.

### 3.4. Indicator Species Based on MRT

From the results of MRT, all sites in the artificial disturbance community were divided into four subgroups using two environment factors (Figure 6A). The error, cross-validation error, and standard error of the model were 0.636, 1.38, and 0.0993, respectively. The sites were divided on the basis of the annual precipitation seasonality (47.28 and 48.78), and the mean temperature of warmest quarter was 27.33 °C. All sites in the natural community were divided into four subgroups using three environmental factors (Figure 6B). The error,

cross-validation error, and standard error of the model were found to be 0.705, 1.46, and 0.121, respectively. The sites were divided by isothermality (27.9), mean temperature of wettest quarter (22.88 °C), and precipitation of wettest month (217.5 mm). The bar represents the distribution of species in four subgroups, and some species were frequently distributed in the specific subgroups, indicating that some species were highly sensitive to the habitat gradient and dominated the division of subgroups. Subsequently, the indicator species in each subgroup were filtered by the indicator value ($p < 0.05$). The significant indicator species of the artificial disturbance community was identified from subgroup III and IV. Group III was *Miscanthus sinensis* (0.98, $p = 0.003$), and group IV was *Dicranopteris dichotoma* + *Indocalamus tessellatus* + *Miscanthus floridulus* (1, 1 and 0.97, $p = 0.01$, 0.01, and 0.012, respectively). The significant indicator species of the natural community were identified from I and III subgroup. Group I was *Phyllostachys heterocycla* (0.92, $p = 0.001$), and group III was *Imperata cylindrica* (0.91, $p = 0.002$).

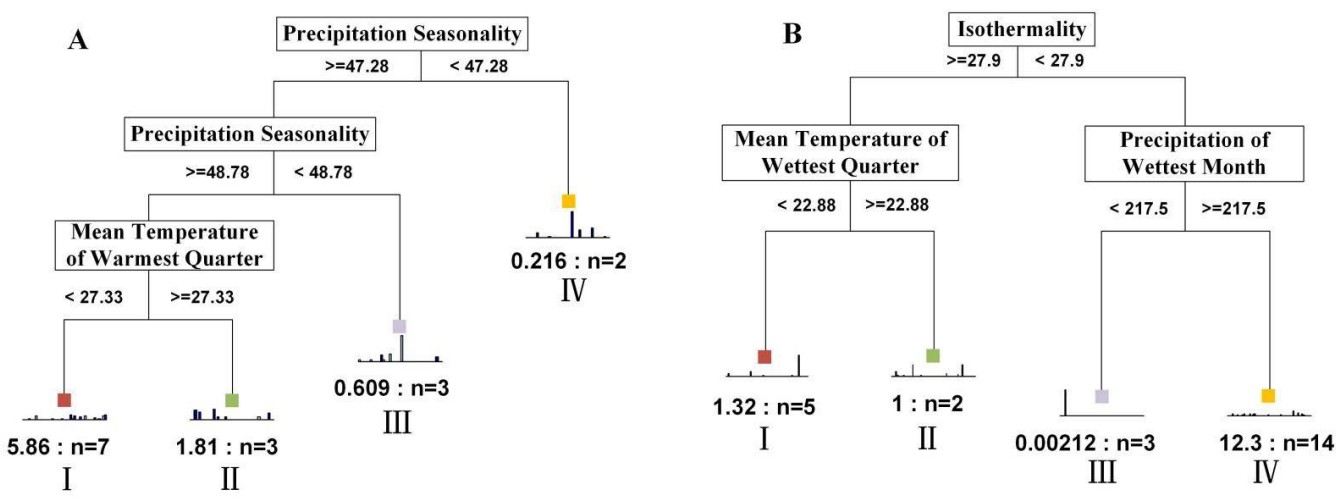

**Figure 6.** Multiple regression tree for classification of artificial disturbance (**A**) and natural community (**B**).

*3.5. Horizontal Distribution Pattern and Formation Mechanism of Diversity*

The linear regression between the diversity index latitude is shown in Figure 7A. The three diversity indices descended with a rise in latitude, and the regression between the diversity index and altitude showed that the species richness of the natural and whole community exhibited a single-peaked curve with an increase in altitude, while the trend of the artificial disturbance community decreased with a rise in altitude (Figure 7B).

Linear regression between the β-diversity and environmental/geographic distance showed that the differences in the species composition increased with the rise in the environmental/geographic distance among the whole, artificial disturbance, and natural communities (Figure 8A–C). Among the whole and natural communities, Mantel test analysis indicated that the β-diversity exhibited a significant positive correlation with both the environmental and geographic distance. Partial Mantel test analysis indicated that the environment had no significant effect on community establishment when the geographic differences were eliminated. However, the geographic factors exhibited a significant effect when the environment differences were eliminated ($p = 0.05$, Table 4). In the artificial disturbance community, Mantel test analysis indicated that the β-diversity exhibited a significant positive correlation only with environmental distance, and the environment had no significant effect when the geographic differences were eliminated. The results showed that both habitat filtering and neutral effect affected the establishment of the whole and natural communities; however, habitat filtering manifested a greater effect. On the

contrary, the establishment of the artificial disturbance community was affected only by habitat filtering.

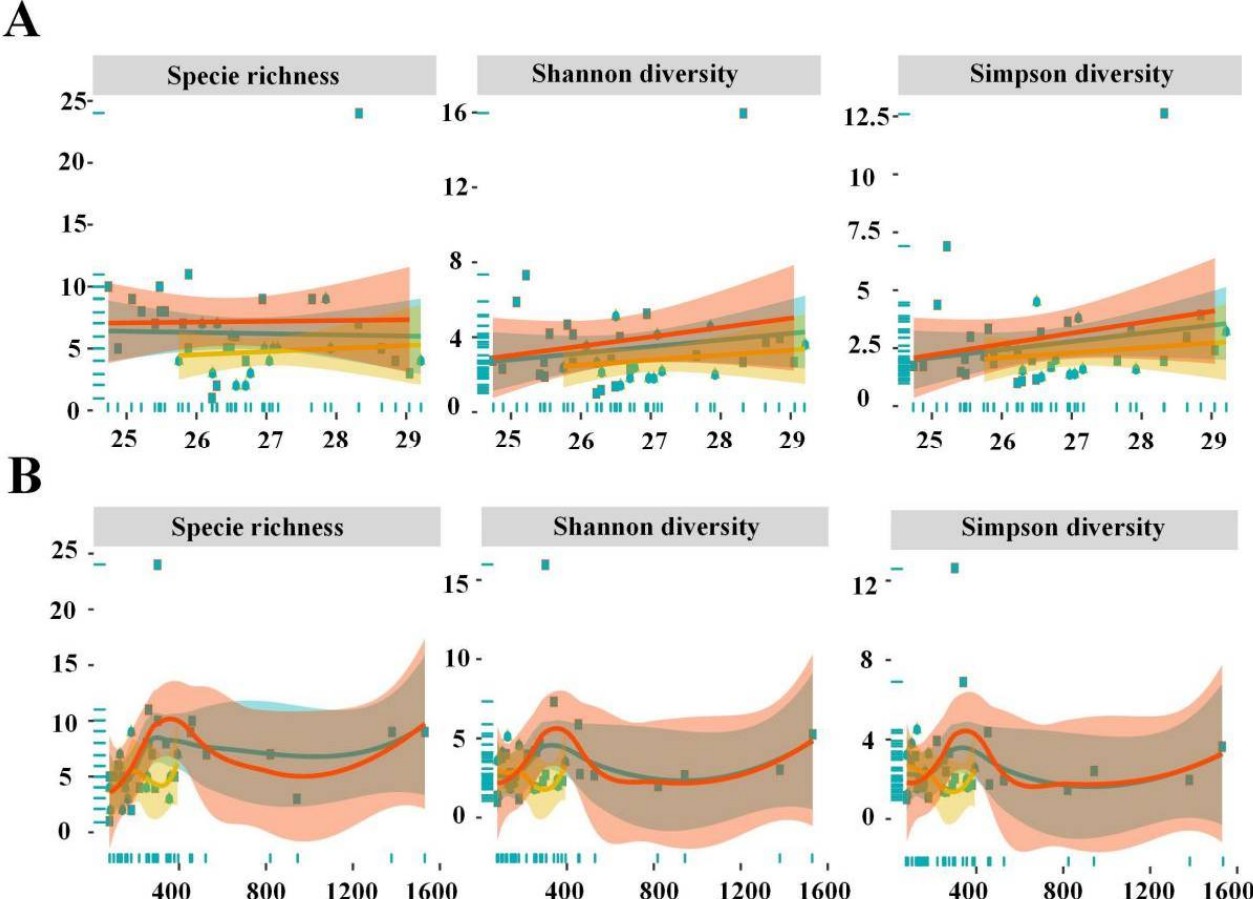

**Figure 7.** Variation in the diversity index along the latitude (**A**) and altitude (**B**) gradients.

**Table 4.** Mantel and partial Mantel test between β-diversity and environment/geographic distance.

|  | Environment Distance | Environment Distance, Eliminate Geographic Distance | Geographic Distance | Geographic Distance, Eliminate Environment Distance | Type |
|---|---|---|---|---|---|
| Statistic r | 0.1998 | 0.06522 | 0.2611 | 0.1522 | All |
| Significance | 0.0072 ** | 0.121 | $8.00 \times 10^{-4}$ *** | 0.001 ** | |
| Statistic r | 0.2611 | 0.2434 | 0.2171 | 0.1564 | Artificial disturbance |
| Significance | 0.0115 * | 0.008 ** | 0.0678 | 0.099 | |
| Statistic r | 0.294 | 0.06308 | 0.3131 | 0.1672 | Natural |
| Significance | 0.0046 ** | 0.206 | 0.0016 ** | 0.015 * | |

'***', '**', and '*' represent significance levels of 0.001, 0.01, and 0.05, respectively.

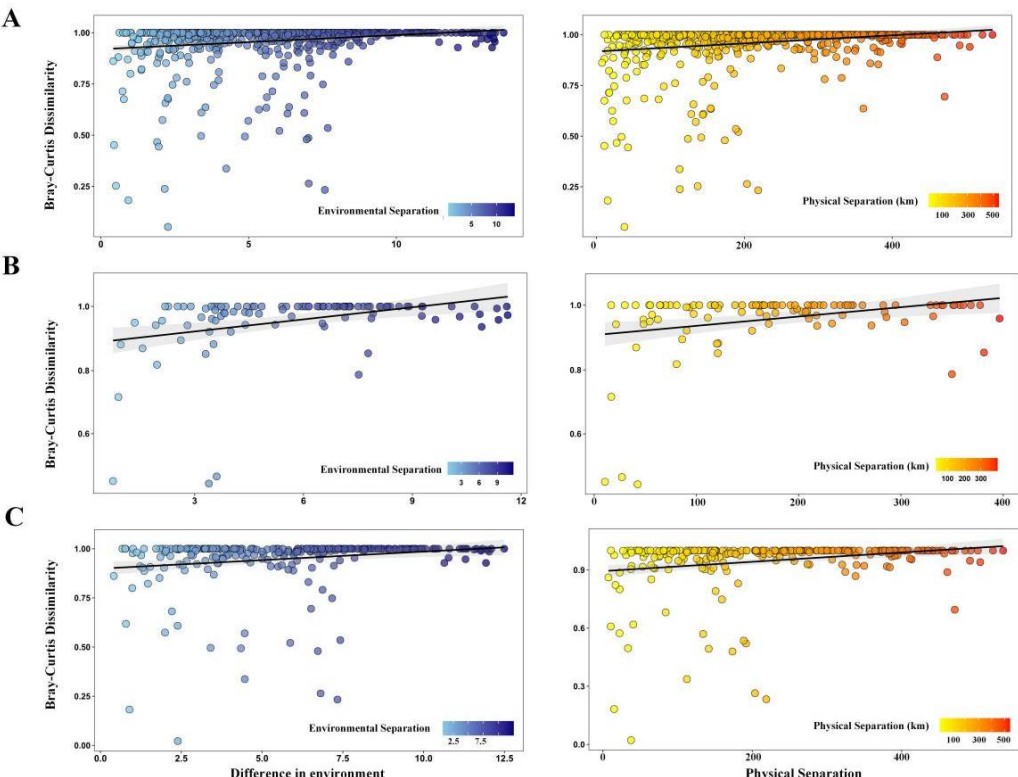

**Figure 8.** Variation in β-diversity along environmental/geographic distance; (**A**) whole community; (**B**) artificial disturbance community; (**C**) natural community.

## 4. Discussion

A majority of the SAD analyses were undertaken at local scales, but less is known at the larger metacommunity scales [5]. Different SADs indicate different ecological processes, and these processes are not mutually exclusive. Hence, it is of great significance to use various models to reveal the relative contributions of the SAD pattern [42]. In this study, the biodiversity of the two types of the shrub communities in Hunan province has been reported for the first time. Although the species richness of the natural community was higher than the artificial disturbance community, the six α-diversity indices showed no significant difference. Previous SAD studies have focused either on finding the best-fitting model given a set of local-scale ecological data or testing the performance of a particular theory or model [42,43]. There has been increasing recognition of the importance of assessing the changes in properties of different SADs across the ecological gradients [6,27]. The broken-stick model is thought to be better in fitting SAD of small homogenous communities with stable population and long life history, while the niche-preemption model is thought to be better in fitting both simple and complex communities [44]. The log-normal model represents a random process, and the Zipf–Mandelbrot model supports hypotheses about the underlying processes linking the requirements of various species with probabilities of encountering the optimal growth conditions in the environment [45]. The fitting results indicated that the SAD followed more than one rule ($p < 0.05$), and the Zipf and Zipf–Mandelbrot models were found to be better than other models. Additionally, the shape parameter of the gambin model was used to characterize the shape of the SAD. The parameter of Zipf–Mandelbrot and gambin revealed interesting relationships between the community evenness, ecological predictability, and environmental diversification. The results indicated that the artificial disturbance community exhibited an unstable structure compared to the whole and natural communities, with higher dominance of the dominant species. We speculate that the reason is that most of the resources and space of the artificial disturbance community was occupied by planted economic species, leading to strong

interspecies competition, with the dominant species having an absolute advantage and the others having little contribution to the community. The natural and whole communities had less interference or larger sampling scale and provided more resource and space, wherein rare species were easier to preserve and reproduce. As compared to the natural community, the frequent immigration events in the artificial disturbance community led to a complex structure with a large number of rare species. This proves that the communities are in the stage of restoration succession. Our results also indicated that the α-diversity index was unable solely to fully reflect the diversity pattern.

The decrease in species richness with an increase in the latitude is a widely accepted phenomenon [46]. Unimodal and monotone decreasing modes are most common in elevation gradients [47]. Previous studies have shown that the strong relationships between climate, topographic, factors and plant richness are mainly impacted by temperature variations as well as latitude and altitude effects [48–51]. This usually produces a mid-domain effect, in which the richness peaks at medium gradients [52], consistent with the intermediate disturbance hypothesis (IDH) and the niche-assembly hypothesis for trees, shrubs, and herbs [53–55]. In the present study, it was found that diversity decreased with an increase in altitude and exhibited a peak value at medium altitude gradients. This indicates that other factors impacted the latitude effect at the regional scale due to the complex terrain of study area, which was dominated by mountains and hills, with uneven distribution of hydrothermal conditions. This leads to the formation of a highly heterogeneous habitat unit, which impacts the distribution of the diversity. Similar results were also found in earlier studies [56–58].

The GBM model aims to verify and compare the contribution of the explanatory variables to the species richness. We found six strongly influencing plant-richness factors: precipitation of wettest month, altitude, annual precipitation, slope, annual mean temperature, and isothermality. Numerous studies have indicated that temperature and precipitation significantly affect the richness patterns [48,59,60]. Slope affects diversity by influencing the sunlight, soil fertility, and soil moisture [19]. Altitude and precipitation of the wettest month exhibited a strong positive relationship with plant richness in our results, but the peaks of diversity occurred within a certain range (Figure 5C,D). The hydrothermal condition indicates energy, and our results suggest that there could be a range of saturation in the use of energy along the elevation gradient before the species with higher energy requirements formed, thereby indicating a redundancy in energy [38].

β-diversity describes the change in the community species composition on temporal and spatial scales [61]. The influence of environmental variables and spatial distance on β-diversity has been a subject of much research in recent years [35,62]. In this study, we evaluated the environment and spatial distance gradients of β-diversity. We applied regression analysis to assess the trend of β-diversity with environmental variables and spatial distance, and conducted variation partitioning to analyze the relative importance of the measured environment and spatial distance on β-diversity. Both the environment and geographic distance exhibited a positive correlation with β-diversity. The trend of whole and natural communities was not steep, but the differences were nevertheless significant. This result is similar to the results of several previous studies [35,61,62]. In the whole and natural communities, the neutral effect played a dominant role in community establishment; however, the establishment of the artificial disturbance community was affected only by habitat filtering. This could be due to the limited resources of the artificial disturbance community.

## 5. Conclusions

Our study reports the biodiversity pattern of the shrub communities in Hunan province for the first time. There was no difference in α-diversity values between the artificial disturbance and natural community. Zipf and Zipf–Mandelbrot models were found to exhibit good SAD fit with the communities, and revealed a different diversity structure. MRT and indicator species analysis identified the domain factors and indicator

species of each community. GBM analysis indicated that the different community structures modulated the utilization efficiency of the vegetation to the environment. β-diversity analysis indicated that the community-establishment mechanism of the natural and artificial disturbance exhibited certain differences. There are some deficiencies in this study, such as the phylogenetic diversity of species and its influencing factors needing to be studied deeply. Nevertheless, our study helps to understand the ecological process of the diversity maintenance in the shrub community and provides theoretical guidance in the restoration process of shrub diversity.

**Supplementary Materials:** The following supporting information can be downloaded at: https://www.mdpi.com/article/10.3390/f13071090/s1, Figure S1: Heatmap of distribution probability of species in plot; Figure S2: Species accumulation curves; Figure S3: Co-occurring network of artificial disturbance community (A) and Natural community (B); Table S1: Predicted species richness of each plots.

**Author Contributions:** Research conceptualization, Y.T. and N.D.; methodology and analysis, N.D. and Q.S.; writing and editing, N.D. and F.M. All authors have read and agreed to the published version of the manuscript.

**Funding:** This research was supported by Forestry Science and Technology Innovation Project of Hunan Province (HNGYL-2019-01 and XLKY202210).

**Institutional Review Board Statement:** Not applicable.

**Informed Consent Statement:** Not applicable.

**Acknowledgments:** The authors wish to thank anonymous reviewers for their constructive reviews.

**Conflicts of Interest:** The authors declare no conflict of interest.

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
