# Peer review of "Patterns and Driving Factors of Diversity in the Shrub Community in Central and Southern China"

_forests, doi:10.3390/f13071090_

Round 1

Reviewer 1 Report

The authors examine the differences between shrub communities of natural character and those subjected to anthropopression, as well as the influence of various environmental and climatic factors on their development and diversity. The knowledge gained may be useful in conservation or restoration work for such communities. However, the article needs significant revision, I have provided detailed comments below. I am not a native speaker, however, I see a lot of problems with the English language used, it would also require additional work on the part of the authors.

Title

I propose to remove the words "Examining" and "potential"

Abstract

L20 – shouldn't this sentence read: " „β diversity analysis indicated that both habitat filtering AND WHAT? had greater effect…”?

Introduction

The first part (paragraph) is not very coherent, it is a conglomeration of several different pieces of information. Please connect them better.

L33 - "mechanism of species assembly rules" - sounds somehow strange (mechanism of rules?)

L41 - "spatial distribution of species diversity" or simply "spatial distribution of species"?

L63 - the word "also" suggests that something was described earlier that also has ecological functions, however, this is not the case

L65-70 - this sentence is much too long and additionally gives the impression of being tangled

At the end of the Introduction, it is worth adding 1-2 sentences on what practical significance this research may have, as well as why it may be interesting for an international reader

Materials and Methods

Subchapter 2.1 - I have some doubts here, which please clarify in the text. To me, the very term 'shrub community' implies that there are only shrubs and no trees. Meanwhile it was based on plots established in woodland. So is this about shrub communities/species growing in forests under trees? Because I assume that "forest fixed sample plots" were established in forests and not outside forests. However, if instead of in forests they were established where there are only shrubs, how these 39 plots were selected from all - this would also need to be described. In addition, in the region such shrub communities are according to the previous description "an important component of plant diversity in this area", so why were other places where shrubs grow (besides "forest plots") not also selected for the study? In this case, are these 39 plots in a large region a sufficient number (and distribution) that the results are not somehow limited?

L78 - what is an "ecological forest"? is a forest not in itself an ecological entity? please explain this term

L84-85 - "the plots are divided into two categories (natural and artificial disturbance community)"? - on what basis was this division made? This should be described in great detail, as the quality of the results depends on it. What did these disturbances consist of? How severe were they? When did they occur? Are the plots still under their influence? Why is this not included in the full analysis of selected plots, since it certainly directly affects shrub communities. It might be worth collating this information in a table for each disturbed plot separately. Additionally, there is no information on how many plots were classified as natural and how many as artificial disturbance.

L85 - superfluous word According

L85-86 - "The shrub species data and stand dynamics factors are extracted respectively" - what data? list missing

Figure 1 - the legend is illegible, the whole Figure can also be enlarged

L93 (and later in the article) - names should be written with a capital letter

L110 - same title of subchapter 2.3 as 2.2

Subchapter 2.3 - if it was only 39 25x40m sites, couldn't a field visit have been done to check completeness of shrub species lists?

L134 - the term Shannon was previously used, it is worth standardising the nomenclature

Results

L174-177 - sentence to be improved (construction and sense)

Figures 2 and 3 - please enlarge, they are not very readable. Title of Figures 2 - please improve the wording: "Biodiversity between..." Figure 3 - maybe break this into two separate Figures? Currently it is completely unreadable. The choice of colours is also not very contrasting - please vary them more

Table 2 - is not very readable due to too narrow columns. The individual names of "Types" could be given as additional rows above the data set of the type, while "Model" could be given as a number in the table (1, 2 etc), and the explanations of these numbers could be given under the table

Subchapter 3.2 - whether there is any confusion between the numbers of Supplementary figures 1 and 2 (looking at the figures placed on pp. 16 and 17)

L195 - this table is missing at the end of the article

Table 3 - same title as Table 2, but different information

L206 - next to 'altitude' should be 'over 20%', not further. Also, as the third factor listed should be Bio 12, not Bio 13 again, and in brackets 'over 15%, not 'over 20%'.

L214-215 - "the maximize diversity value appeared in S and W-S slope" - is there no error here?

Figure 4 - too small, illegible

L233-237 - please correct the numbering of subgroups (it should be the same as in Figure 5)

L236 - here should be rather natural community

Figure 5 - please enlarge, it is unreadable

Table 3 - duplicates information in the text above Figure 5. Just add p-values there and delete Table 3

L243-244 - "The three diversity index reduced with longitude'rising" - the sentence before and in Figure 6A is LATITUDE, not LONGITUDE (the same error is probably in Abstract). Additionally, it seems to me that in the graphs for Shannon diversity and Simpson diversity the lines are trending upwards, not downwards, with latitude rising

Figure 6 - completely unreadable, too small and colours overlap

Figure 7 - is wrongly numbered (it is 6, and should be 7). Additionally, sets A, B and C are IDENTICAL, they do not differ!

Discussion

Would be good to make stronger reference to anthropogenic disturbance in the part of plots.

L272 - rather "is" than "in"

L276 - rather: two TYPES of shrub communities

L276-278 - "Although the species richness of natural community diversity was higher than artificial disturbance community, but other 6 α diversity index had no significant difference." - no comment on this result. Additionally L277 - redundand word "diversity"

L295-296 - "The reason is that most resources and space of artificial disturbance community are occupied by planted economic species," - do I understand correctly that it was shrubs that were planted and not trees in the forest where these shrubs grow? shrubs are also economically important in this region? if so, it might be worth adding how

L303 - verify against own results (Fig 6A, also in L310). Throughout the sentence, emphasise more what is your own result and what is a statement from the literature

L311 - latitude or altitude?

L317 - influenced or influencing?

L318 -" and so on" - this is not precise - please specify which factors

Conclusions

L341 - "of two shrub communities of this area" - please recall here in more detail what these two main types of shrub communities are and in which region they were surveyed. In general, please break the first sentence into two and add in the second sentence some text before the phrase "6 alpha ..."

L350 - forest or shrubs diversity? in this respect the article is imprecise, please clarify this thoroughly. Shrubs alone do not make a forest, a forest must contain trees (at least in Poland...) - unless it is some kind of transitional phase between one used stand and the next one that is planted in its place.

Supplementary figure 1

I propose to turn 90 degrees and enlarge it to the whole page, it is not very readable at the moment

Reviewer 2 Report

Determination of species diversity is the most fundamental topics in both biology and ecology, it’s important for biodiversity conservation, planning of nature reserves and forest management. Therefore, the topic of paper is relevant.

 The scientific novelty of the study consists in explore the diversity pattern of shrub community, and compare the differences between natural and artificial disturbance community, detect the domain factors that affect the diversity pattern, reveal the relative importance of habitat filters.

 The study will help understanding the ecological process of shrub community diversity maintenance, and provide theoretical guidance for the restoration process of forest diversity.

 In the introduction, the relevance of the study is well substantiated and the state of the problem is described. The research objectives are formulated clearly and clearly. However, there is a lack of information on the status of studies on SADs over the past 2 years. Perhaps the authors will be interested in the following publications:

https://doi.org/10.1051/e3sconf/202125402003

http://dx.doi.org/10.12775/EQ.2021.011

 The research area is practically not described. The authors provide only a map. It is desirable to describe in detail the natural conditions, relief, soil, vegetation in accordance with the accepted classification systems of it.

 The methodology is described in detail. The authors used modern methods of analysis. The choice of methods is reasonable and adequate for the tasks set. The forest survey data were obtained from forest fixed sample plot investigation database of Human ecological forests (update in 2019). Additionally, 19 bioclim variables of each site were retrieved from the world climate database (WorldClim: http://www.worldclim.org/).

Species diversity (α diversity) was assessed using the most commonly used diversity indices: richness, shannon, shannon entropy, shannon’s evenness (Hill ratio), simpson, and Simpson’s evenness (Hill ratio) and Pielou’s evenness, pairwise comparison is performed with Wilcoxon test. Rank abundance dominance (RAD) plots display logarithmic species abundances against species rank order to analysis types of abundance distributions. In this study, 5 models were used, i.e., brokenstick model, niche preemption model, log-normal model, Zipf model and Zipf-Mandelbrot model.

 The research results are perfectly illustrated with figures and tables that are informative and do not duplicate each other.

It is necessary to correct an error in the numbering of the figures: two different figures are designated as figure 6.

 Conclusions follow from the results and are reasonable.  The article will be of interest to a wide range of readers whose scientific interests are related to ecology, forestry, as well as climate change. Despite the fact that English is not my native language, I read the paper with interest and had no difficulties in understanding. The paper fully corresponds to the subject and level of the Forests.

However, it would be interesting if the authors point out unresolved issues and outline directions for further research.

Author Response

Please find the revised manuscript entitled "The patterns and drivers of shrub community diversity in Central and Southern China" (Manuscript ID: 1775378) here attached. The authors are grateful for the valuable and helpful comments of the reviewers. According to the comments, we have made significant improvements of the manuscript as follows. Additionally, our manuscript have underwent extensive English revisions with track changes mode in the manuscript.

The research area is practically not described. The authors provide only a map. It is desirable to describe in detail the natural conditions, relief, soil, vegetation in accordance with the accepted classification systems of it.

Answer:Thanks for your valuable suggestion, we have added the description of study area in line69-73.

It is necessary to correct an error in the numbering of the figures: two different figures are designated as figure 6.

Answer:Thanks for your valuable suggestion, we have revised it.

However, it would be interesting if the authors point out unresolved issues and outline directions for further research.

Answer:Thanks for your valuable suggestion, we have revised it in conclusions.

Round 2

Reviewer 1 Report

The article has been largely improved. It is very inconvenient that only the linguistic corrections are made visible, while the substantive ones are not. The second version of the article uploaded by the authors also lacks line numbering, which makes it difficult to review and write comments on the article. I provide detailed comments below.

Introduction

Still I think, that at the end of the Introduction, it is worth adding 1-2 sentences on what practical significance this research may have, as well as why it may be interesting for an international reader

Materials and Methods

Subchapter 2.1 –  I propose to add the first sentence that the research was conducted in an ecological forest - because the current first sentence seems little related to the subsection "Data Sources" (I propose to move it further, behind the word "province"). Please also add information on whether timber is harvested in these forests, as this is a factor that also affects shrubs communities (e.g. through light input).

Text above Figure 1 - "the shrub species data" - please add which data? only the list of species or also their abundance etc.?

Figure 1 - I think I misspelled previously what needs to be improved. So I will explain again: the previous layout of both maps was good, better than the current one, so please go back to it. All that was needed was to enlarge the WHOLE Figure (stretching the outer frame) and to enlarge the legend to the plots map, because it is unreadable (especially in the potential printed version)

Subchapter 2.2 (and 2.4) - "Richness" is unnecessarily capitalized (it's not a surname). I propose to give as it was before (species richness)

Subchapter 2.4 - Curtis should be capitalized (Bray-Curtis distance)

Results

Table 2 - in the first line it is worth stating "All plots" instead of "Plots", it will be clearer. Footnotes - should be M1-M5, not M1-M2

Figures 3 and 4 - despite the enlargement they are still not very legible - blurred and with too small font

Subchapter 3.2 - there are references to supplementary table and figures, while the version of the article I received for review does not contain these materials, nor does References

Subchapter 3.3 - at the beginning there should be a reference to Figure 5A, not Figure 4A

Figure 5 - - too small, illegible. Is the word 'plots' correctly used in the title for D?

Subchapter 3.4 - still in one place is group 3 and group 4, instead of group III and group IV

Figure 6 - please enlarge, it is unreadable

Subchapter 3.5 - the first line should probably read "between the diversity index AND latitude". Whereas in the second sentence it says "The three diversity index reduced with a rise in the latitude" - is this true for Shannon and Simpson diversity (Figure 7A)?

Figure 7 - still unreadable

Mantel should be capitalised throughout this subchapter

Discussion

Should be (Figure 5C, D), not 4C, D

Should be: This indicates that other factors impact the latitude altitude effect at the regional scale, and may causeddue to by the complex terrain of study area , which dominated by mountains and hills, with uneven distribution of hydrothermal conditions.

Author Response

I am very sorry that I uploaded the wrong version, and I have uploaded the Correct version. The response of ropund 1 and round 2 are listed as follows:

Round 1

Title

I propose to remove the words "Examining" and "potential"

Answer:Thanks for your valuable suggestion, we have removed the "Examining" and "potential" in the title (line 2-3).

Abstract

L20 – shouldn't this sentence read: " „β diversity analysis indicated that both habitat filtering AND WHAT? had greater effect…”?  

Answer:Thanks for your valuable suggestion, we have revised this sentence in line 21.

Introduction

The first part (paragraph) is not very coherent, it is a conglomeration of several different pieces of information. Please connect them better.

Answer:Thanks for your valuable suggestion, we have revised this paragraph in line31-43.

L33 - "mechanism of species assembly rules" - sounds somehow strange (mechanism of rules?)

Answer:We have revised this to “species assembly rules” in line40.

L41 - "spatial distribution of species diversity" or simply "spatial distribution of species"?

Answer:We have revised this to “spatial distribution of species” in line48.

L63 - the word "also" suggests that something was described earlier that also has ecological functions, however, this is not the case

Answer:We have deleted the word in line 70.

L65-70 - this sentence is much too long and additionally gives the impression of being tangled

Answer:We have revised this sentence in line 72-77.

At the end of the Introduction, it is worth adding 1-2 sentences on what practical significance this research may have, as well as why it may be interesting for an international reader

Answer:We have added relevant sentences in line85-88 .

Materials and Methods

Subchapter 2.1 - I have some doubts here, which please clarify in the text. To me, the very term 'shrub community' implies that there are only shrubs and no trees. Meanwhile it was based on plots established in woodland. So is this about shrub communities/species growing in forests under trees? Because I assume that "forest fixed sample plots" were established in forests and not outside forests. However, if instead of in forests they were established where there are only shrubs, how these 39 plots were selected from all - this would also need to be described. In addition, in the region such shrub communities are according to the previous description "an important component of plant diversity in this area", so why were other places where shrubs grow (besides "forest plots") not also selected for the study? In this case, are these 39 plots in a large region a sufficient number (and distribution) that the results are not somehow limited?

L78 - what is an "ecological forest"? is a forest not in itself an ecological entity? please explain this term

Answer:Ecological forest is divided artificially, which has high ecological and social benefits, ecological forest has the function of maintaining ecological balance and protecting biodiversity. Among them, the forest with shrub coverage greater than or equal to 40% is identified as shrub communities. The formation of shrub communities are caused by deforestation, natural disasters, planting activities and so on. We have revised in line95-115.

L84-85 - "the plots are divided into two categories (natural and artificial disturbance community)"? - on what basis was this division made? This should be described in great detail, as the quality of the results depends on it. What did these disturbances consist of? How severe were they? When did they occur? Are the plots still under their influence? Why is this not included in the full analysis of selected plots, since it certainly directly affects shrub communities. It might be worth collating this information in a table for each disturbed plot separately. Additionally, there is no information on how many plots were classified as natural and how many as artificial disturbance.

Answer:We have revised in line 95-107.

L85 - superfluous word According

Answer:We have deleted the word.

L85-86 - "The shrub species data and stand dynamics factors are extracted respectively" - what data? list missing

Answer:We have add the stand dynamics factors in line114-115.

Figure 1 - the legend is illegible, the whole Figure can also be enlarged

Answer:We have revised the figure and legend in line118-119.

L93 (and later in the article) - names should be written with a capital letter

Answer:We have revised this in line 115-116.

L110 - same title of subchapter 2.3 as 2.2

Answer:We have revised this in line 140.

Subchapter 2.3 - if it was only 39 25x40m sites, couldn't a field visit have been done to check completeness of shrub species lists?

Answer:All the species and its number are from field surveys and completed by forestry departments from each county. The shrub species in other forest type are analyzed in other studies, and not included in this research.

L134 - the term Shannon was previously used, it is worth standardising the nomenclature

Answer:Thanks for your valuable suggestion, we have revised in whole article.

Results

L174-177 - sentence to be improved (construction and sense)

Answer:Thanks for your valuable suggestion, we have revised in line203-206. 

Figures 2 and 3 - please enlarge, they are not very readable. Title of Figures 2 - please improve the wording: "Biodiversity between..." Figure 3 - maybe break this into two separate Figures? Currently it is completely unreadable. The choice of colours is also not very contrasting - please vary them more

Answer:Thanks for your valuable suggestion, we have remade the figure (break figure 3 into two separate Figures) and improved the legend.

Table 2 - is not very readable due to too narrow columns. The individual names of "Types" could be given as additional rows above the data set of the type, while "Model" could be given as a number in the table (1, 2 etc), and the explanations of these numbers could be given under the table

Answer:Thanks for your valuable suggestion, we have remade the table.

Subchapter 3.2 - whether there is any confusion between the numbers of Supplementary figures 1 and 2 (looking at the figures placed on pp. 16 and 17)

Answer:Thanks for your valuable suggestion, we have renumber the Supplementary figures .

L195 - this table is missing at the end of the article

Answer:Thanks for your valuable suggestion, the missing table was put at the end of the article.

Table 3 - same title as Table 2, but different information

Answer:Thanks for your valuable suggestion, we have revised the table and table legend.

L206 - next to 'altitude' should be 'over 20%', not further. Also, as the third factor listed should be Bio 12, not Bio 13 again, and in brackets 'over 15%, not 'over 20%'.

Answer:Thanks for your valuable suggestion, we have revised this in line244.

L214-215 - "the maximize diversity value appeared in S and W-S slope" - is there no error here?

Answer:Thanks for your valuable suggestion, this sentence is is inappropriate here, we have deleted it.

Figure 4 - too small, illegible

Answer:Thanks for your valuable suggestion, we will upload the figure with higher pixel.

L233-237 - please correct the numbering of subgroups (it should be the same as in Figure 5)

Answer:Thanks for your valuable suggestion, we have revised the numbering of subgroups in line260-263 .

L236 - here should be rather natural community

Answer:Thanks for your valuable suggestion, we have revised this in line275.

Figure 5 - please enlarge, it is unreadable

Table 3 - duplicates information in the text above Figure 5. Just add p-values there and delete Table 3

Answer:Thanks for your valuable suggestion, we have revised this.

L243-244 - "The three diversity index reduced with longitude'rising" - the sentence before and in Figure 6A is LATITUDE, not LONGITUDE (the same error is probably in Abstract). Additionally, it seems to me that in the graphs for Shannon diversity and Simpson diversity the lines are trending upwards, not downwards, with latitude rising

Answer:Thanks for your valuable suggestion, we have revised this in line269,270,273. The slope of Shannon diversity and Simpson diversity was all positive (all plots: 0.019336308, 0.021680357,0.020811826; Artificial disturbance plots: 0.043172017, 0.045382148, 0.043968062; natural plots: 0.023029573, 0.021599563, 0.02243502).

Figure 6 - completely unreadable, too small and colours overlap

Answer:Thanks for your valuable suggestion, we have remade the figure.

Figure 7 - is wrongly numbered (it is 6, and should be 7). Additionally, sets A, B and C are IDENTICAL, they do not differ!

Answer:Thanks for your valuable suggestion, we have remade the figure.

Discussion

Would be good to make stronger reference to anthropogenic disturbance in the part of plots.

L272 - rather "is" than "in"

Answer:Thanks for your valuable suggestion, we have revised it in line313.

L276 - rather: two TYPES of shrub communities

Answer:Thanks for your valuable suggestion, we have revised it in line323.

L276-278 - "Although the species richness of natural community diversity was higher than artificial disturbance community, but other 6 α diversity index had no significant difference." - no comment on this result. Additionally L277 - redundand word "diversity"

Answer:Thanks for your valuable suggestion, we have revised it in line344-346.

L295-296 - "The reason is that most resources and space of artificial disturbance community are occupied by planted economic species," - do I understand correctly that it was shrubs that were planted and not trees in the forest where these shrubs grow? shrubs are also economically important in this region? if so, it might be worth adding how

Answer:Thanks for your valuable suggestion, all the artificial disturbance community are occupied by economical shrubs (tea-oil tree and citrus). And we add the information in Subchapter 2.1.

L303 - verify against own results (Fig 6A, also in L310). Throughout the sentence, emphasise more what is your own result and what is a statement from the literature

Answer:Thanks for your valuable suggestion, we have revised it in line337.

L311 - latitude or altitude?

Answer:Should be altitude (line351).

L317 - influenced or influencing?

Answer:Should be influencing (line62).

L318 -" and so on" - this is not precise - please specify which factors

Answer:Thanks for your valuable suggestion, we have revised it in line364.

Conclusions

L341 - "of two shrub communities of this area" - please recall here in more detail what these two main types of shrub communities are and in which region they were surveyed. In general, please break the first sentence into two and add in the second sentence some text before the phrase "6 alpha ..."

Answer:Thanks for your valuable suggestion, we have revised it in line388-391.

L350 - forest or shrubs diversity? in this respect the article is imprecise, please clarify this thoroughly. Shrubs alone do not make a forest, a forest must contain trees (at least in Poland...) - unless it is some kind of transitional phase between one used stand and the next one that is planted in its place.

Answer:Thanks for your valuable suggestion, the shrub in ecological forest are thought to be a stage of forest succession, and was managed as forest. And we clarified this in Subchapter 2.1.

Supplementary figure 1

I propose to turn 90 degrees and enlarge it to the whole page, it is not very readable at the moment

Answer:Thanks for your valuable suggestion, we have revised it.

Round 2

Introduction

Still I think, that at the end of the Introduction, it is worth adding 1-2 sentences on what practical significance this research may have, as well as why it may be interesting for an international reader

Answer:We have added relevant sentences in line85-88 .

Materials and Methods

Subchapter 2.1 –  I propose to add the first sentence that the research was conducted in an ecological forest - because the current first sentence seems little related to the subsection "Data Sources" (I propose to move it further, behind the word "province"). Please also add information on whether timber is harvested in these forests, as this is a factor that also affects shrubs communities (e.g. through light input).

Answer:We have added relevant sentences in line95. And the deforestation is not allowed in ecological forest (line99).

Text above Figure 1 - "the shrub species data" - please add which data? only the list of species or also their abundance etc.?

Answer:We have revised it in line116.  

Figure 1 - I think I misspelled previously what needs to be improved. So I will explain again: the previous layout of both maps was good, better than the current one, so please go back to it. All that was needed was to enlarge the WHOLE Figure (stretching the outer frame) and to enlarge the legend to the plots map, because it is unreadable (especially in the potential printed version)

Answer:We have replaced it.  

Subchapter 2.2 (and 2.4) - "Richness" is unnecessarily capitalized (it's not a surname). I propose to give as it was before (species richness)

Answer:We have revised it in line125, 173.  

Subchapter 2.4 - Curtis should be capitalized (Bray-Curtis distance)

Answer:We have revised it in line175.  

Results

Table 2 - in the first line it is worth stating "All plots" instead of "Plots", it will be clearer. Footnotes - should be M1-M5, not M1-M2

Answer:We have revised it in line219.  

Figures 3 and 4 - despite the enlargement they are still not very legible - blurred and with too small font

Answer:We have revised it.  

Subchapter 3.2 - there are references to supplementary table and figures, while the version of the article I received for review does not contain these materials, nor does References

Answer:We have revised it at the end of article.  

Subchapter 3.3 - at the beginning there should be a reference to Figure 5A, not Figure 4A

Answer:We have revised it in line248.  

Figure 5 - - too small, illegible. Is the word 'plots' correctly used in the title for D?

Answer:We have revised it.  

Subchapter 3.4 - still in one place is group 3 and group 4, instead of group III and group IV

Answer:We have revised it in line278.  

Figure 6 - please enlarge, it is unreadable

Answer:We have revised it.

Subchapter 3.5 - the first line should probably read "between the diversity index AND latitude". Whereas in the second sentence it says "The three diversity index reduced with a rise in the latitude" - is this true for Shannon and Simpson diversity (Figure 7A)?

Answer:this should be :The three diversity index descended with a rise in the latitude. (line288)

Figure 7 - still unreadable

Answer:We have revised it.

Mantel should be capitalised throughout this subchapter

Answer:We have revised it in 301, 305, 312

Discussion

Should be (Figure 5C, D), not 4C, D

Answer:We have revised it in 376

Should be: This indicates that other factors impact the latitude altitude effect at the regional scale, and may caused due to by the complex terrain of study area , which dominated by mountains and hills, with uneven distribution of hydrothermal conditions.

Answer:We have revised it in line364-365.

Reviewer 2 Report

The revised version is presented without line numbering and a list of references. Therefore, it is difficult to write a review. Please fix it.
Also I can't rate the items: Does the introduction provide sufficient background and include all relevant references? Are all the cited references relevant to the research? Did you detect inappropriate self-citations by authors?

Author Response

I am very sorry that  I uploaded the wrong version, and we have uploaded the correct version. The response was both listed as follows:

 The research area is practically not described. The authors provide only a map. It is desirable to describe in detail the natural conditions, relief, soil, vegetation in accordance with the accepted classification systems of it.

Answer:Thanks for your valuable suggestion, we have added the description of study area in line95-117.

It is necessary to correct an error in the numbering of the figures: two different figures are designated as figure 6.

Answer:Thanks for your valuable suggestion, we have revised it.

However, it would be interesting if the authors point out unresolved issues and outline directions for further research.

Answer:Thanks for your valuable suggestion, we have revised it in line403-405.